# The Sustainable Effect of Artificial Intelligence and Parental Control on Children's Behavior While Using Smart Devices' Apps: The Case of Saudi Arabia

**Othman Alrusaini** [1,*] and **Hasan Beyari** [2]

1 Department of Engineering and Applied Sciences, Applied College, Umm Al-Qura University, Makkah 24382, Saudi Arabia
2 Department of Administrative and Financial Sciences, Applied College, Umm Al-Qura University, Makkah 24382, Saudi Arabia; hmbeyari@uqu.edu.sa
* Correspondence: oarusaini@uqu.edu.sa

**Abstract:** The study was necessitated by the unprecedented consumption of smart devices by children in Saudi Arabia, which has been a concern to parents and other stakeholders. It investigated the way that game apps, social media apps, and video-streaming apps impact child development. It also examined the roles played by artificial intelligence control and parental control in enhancing the sustainability of children's behavior amidst smart technologies. The theories underpinning this research were the theory of reasoned action and the theory of planned behavior. The population of the study was 1,616,755 households in the eastern, western, and central regions. The researcher used an online survey to capture the sentiments of a sample of 415 parents who had given their children at least one smart device. The primary questionnaire focused on game apps, social media apps, video-streaming apps, artificial intelligence control, parental control, and the sustainability of child behavior. On the other hand, a separate questionnaire designed specifically to capture demographic information was also drafted. The structural equation model (SEM) was preferred, as it depicted the moderating roles of artificial intelligence control and parental control by using SPSS AMOS software. Findings established that games, social media, and video-streaming applications negatively affected the sustainability of child behavior. The findings presented in this paper show that the moderating effect of artificial intelligence control was more statistically significant than parental controls in influencing the sustainability of child behavior. Moreover, the results show that the greatest effect on children's behavior were social media, video-streaming, and games apps. respectively. Nevertheless, both approaches resulted in positive child behavior. Hence, the study concluded that using artificial intelligence control is more effective than relying on parental controls to enhance the behavioral sustainability of children with smart device applications in Saudi Arabia.

**Keywords:** children's behavior; artificial intelligence control; parental control; games apps; social media content; video-streaming apps; SEM; Saudi Arabia



## 1. Introduction

Two decades ago, it was unimaginable that children would be owning and using mobile devices similar to adults. The landscape has dramatically changed, as 35.6% of children age between 4 and 14 years old now spend between one and two hours on their mobile devices, while those spending more than four hours are now 15.1% [1]. Even more critical is the age at which children receive their first smartphones. Statistics suggest that most children receive their first mobile device between ages 11 and 12 [1]. At this age, they are primarily still within their prepubescent years. Their exposure to these gadgets at an early age has been a subject of concern to their parents, who feel that they cannot handle the nature of some online material. Their specific fear is that their children will come

into contact with some age-inappropriate content, which is not good for the sustainable development of their mental health [2].

In Saudi Arabia, the earliest age children get their first mobile device is now just seven. Parents are becoming more comfortable with their children owning these devices, as 60% of Saudi parents find the devices helpful in fostering their children's problem-solving skills [3]. Another 64% of parents find the devices helpful in instilling a sense of responsibility in children. These statistics notwithstanding, about 53% of the parents argue that these devices have caused sleep difficulties and other attention-related challenges to their children [3]. The sustainability of continued exposure to digital devices among children concerns local and international children-related bodies. United Nations Children's Fund (UNICEF) raises concerns that Internet access by children continues to be a more personal and confidential matter for them, as most of them prefer to use these devices in their bedrooms, with limited supervision [4].

The subject of sustainable child development and behavior in a digital world has attracted mixed opinions. Proponents of digital integration of children contend that exposing children to technology at an early age is beneficial to developing their academic and technological curiosity [5,6]. Some even argue that this technology further facilitates educational progress by exposing children to volumes of valuable materials. As intimated in the findings above, scholarly opinion is split between allowing children limited access to mobile devices and completely disallowing this access. The issue with disallowing access is that the children would find other means of accessing the materials all the same. It is far more dangerous because parents would not be in control of this access, and the material could be more inappropriate. Children often model their behavior from the materials they interact with online, and it is an unsustainable practice for them to have unlimited access to online materials [7].

This issue has been a subject of debate in technology circles where content, games, and social media owners continuously embrace the fact that children consume their products. Controlling this access is a big challenge because it is difficult to tell whether the user is a child or an adult. However, video-streaming service companies such as YouTube and Netflix have special kids platforms. YouTube runs a separate program known as YouTube Kids, which has content specifically audited for children [8]. They mostly run on artificial intelligence (AI), which is a branch of technology that builds models to emulate and predict human behavior [9]. Apart from these AI-based controls, parental control also manifests. It is in the form of parents paying close attention to the websites and content their children have been interacting with online. Unfortunately, parents' engagement in actively inspecting children's online activities is minimal. An estimated three in four parents either do not find time to involve themselves with their children's online activities or are unaware that their children have active online lives [10]. This unsupervised access to limitless online materials, games, and content may result in the unsustainable development of children's mental and psychological health [11].

The general objective of the paper is to establish the effect of various digital media interfaces on child development and how artificial intelligence control and parental controls moderate this effect. Specifically, the paper seeks to achieve the following objectives:

1.  To determine the effect of game apps on the sustainability of children's behavior in Saudi Arabia;
2.  To examine the significance of the effect of video-streaming apps on the sustainability of children's behavior in Saudi Arabia;
3.  To analyze the effect of video-streaming apps on the sustainability of children's behavior in Saudi Arabia;
4.  To establish the moderating effect of artificial intelligence control and parental controls on the effect of digital media interfaces on the sustainability of children's behavior in Saudi Arabia.

This paper is significant because of its implication for children's growth. Indeed, several studies have investigated the effect of digital media interfaces on children's behavior.

However, most of them fail to appreciate the role played by artificial intelligence control and parental controls in alleviating the negative effects of their technological exposure. The research is progressively innovative because it seeks to provide AI-based solutions to the growing problem of children's inappropriate exposure to online content. The researcher argues that physical controls are highly limited in their moderating role, thereby creating the need to implement more automated measures.

The paper is organized into six sections. Section 1 introduces the study by providing an elaborate background, objectives, and research contribution. Next, the paper provides a comprehensive review of literature materials to determine the state of research on the subject. In the theoretical background section, the researcher examines two relevant theories, draws up the conceptual framework, and develops hypotheses. The paper proceeds to formulate the methodology of the study before presenting and analyzing the findings. Ultimately, the researcher discusses these findings in light of reviewed literature, concludes, and makes recommendations on how best to leverage artificial intelligence (AI) and parental controls.

## 2. Literature Review

The use of game apps among children has served to mold the sustainability of their behavior in the real world. Some teachers use serious games to create a sense of attentiveness among young students [12]. Consequently, these games create a culture of attentiveness among children by enhancing their gaze durations, which results in better-sustained attentiveness. Some scholars argue that violent game applications are retrogressive in advancing sustainable child behavior.

A game like Fortnite has been at the receiving end of criticism for exposing children to violent behavior. The game depicts cartoonish characters that go shooting each other [13]. The goal is for the player to be the last one standing. Research has rendered these games ineffective in explaining child development and behavior [14]. If played in a multi-player setting, the games may even help develop prosocial behavior. Some of the critical social qualities gained by children as they play Fortnite in this setting are cooperation, resilience, self-expression, and the appreciation of others' successes [14]. Children satisfy key social needs such as positive emotions, enjoyment, and relatedness by participating in gaming applications. All these factors contribute positively to sustainable development during the early childhood phases.

Children's use of social media may make some parents feel uncomfortable because of the unregulated nature of most of the content posted on these platforms. Nevertheless, adolescents keep flocking to these sites, where some may even use social media accounts that are not theirs [15]. In other cases, parents may open Facebook and Instagram accounts for their children while they are still young. The current TikTok craze has made many children flock to the platform. The effects of interacting with these sites are well-documented in scholarly research. There are divided opinions on how these platforms affect children's sustainable development, with some indicating that it causes depression [16]. Some have shown that children interacting with others on social media sites are at risk of developing queer behavior [17]. This adverse behavior often results from the children becoming curious and engaging with people unknown to them. Some users on the sites are adults masquerading as teens or youths. Many of them have sinister motives, such as child trafficking or pedophilia. Other sources claim that social media engagement is important to a child's sustainable development because it accords them with an environment to exercise their social skills [18]. Children on social media also learn a great deal from their peers, who come from various places around the globe. This networking aspect of social media is vital to the child's embracing the global village.

Children find video content highly attractive, especially if it is animated because it appeals to their simplicity. Unfortunately, not all online video content is appropriate for children, which is not good for their sustainable behavioral development [19]. A site such as YouTube has regular and kids' versions. Some children are too curious to remain locked in

the kids' zone, and they explore the regular site. YouTube's attempts to moderate content on the platform are commendable, as it is difficult to come across inappropriate content if one does not search for it [20]. However, children are extremely curious, and in an unsupervised environment, they may search for content that is not appropriate for their ages. From this angle, it may seem that children's usage of video-streaming services is detrimental to their behavioral conduct. It is worth noting that the sites play critical developmental roles in children's lives by allowing them to view content that promotes their social well-being. Most of the content hosted on YouTube Kids and Netflix Kids teaches moral lessons to children, which is good for their sustainable development [21]. Some children may even extrapolate the lessons learned in these videos to their real-life experiences.

To mitigate the murky waters of children's online experiences, some have reached out to algorithmic solutions in the form of artificial intelligence. It is a computer science technology that trains and builds models to emulate human behavior [9]. Normally, it would take a human being to determine whether the content is appropriate for viewers or users. It is mostly applicable in video content recommendation. Using the predictive power of machine learning AI and data obtained from a user's watch history, the algorithm can determine how to populate search results and recommended videos [22]. Children are likely to view and watch content appropriate to their ages. Hence, the system can detect their age with a small margin of error. After this determination, it profiles the user as a child and recommends the most relevant content. If the device is in use by adults and children, it may confuse the algorithm and result in less accurate predictions [23]. Some social media sites have integrated algorithms to detect unusual usage in an account. If such is the case, the system may lock the user out and prompt them to enter their security information. This mitigation technique is useful when a child logs in to an adult's social media account.

Machine learning (ML) technologies classify content based on features and user feedback. Whenever a user uploads an image or video to a social media site (Facebook) and video-streaming platforms (YouTube), a bot scans this image or video frame using ML tools such as OpenCV to determine whether the content is graphic or in any way inappropriate. In this way, it is extremely hard to find inappropriate materials on these sites [24]. The bot also scans videos posted and blurs their thumbnails. Some social media platforms such as Twitter further provide a warning to users that the content is graphic and may be inappropriate to some audiences [25]. In text-based classification, social media sites have a big data dictionary of offensive phrases. Whenever one makes a post containing any such phrases, the bot replaces the offensive part with asterisks.

Even in the presence of artificial intelligence to control children's exposure to inappropriate content, some still consider parental controls an even more suitable approach. Such controls may be manual or embedded in the systems. Manual controls involve parents physically supervising the content with which their children interact [19]. Some choose to download what they find appropriate and switch off the Internet so that children only consume what they have proofed. Others choose to be close to the children while playing games, browsing social media sites, and watching online videos to ensure they are within their limits [26]. Even with the manual nature of most parental controls, a significant proportion of parents still implement them [26]. Such parents consider their physical presence intimidating enough to influence their children's online conduct.

## 3. Theoretical Background

### 3.1. Theory of Reasoned Action

The theory of reasoned action (TRA) explains people's conduct based on their intention to behave. The intention is the greatest indicator of whether individuals accomplish anything [27]. The desire to do an activity is impacted by a person's attitude toward it, the attitudes of key individuals, and perceived societal pressures, otherwise known as subjective norms. The TRA tried to find social forces that may predict behavior. In modeling children's behavior, proponents of the theory argue that the social environment in which these children live determines their ultimate behavior. In this information age,

they spend most of their social lives on digital platforms. Hence, these interactions act as social pressures that mold their conduct. A child that has seen too much violence online may be insensitive to such acts [28]. In addition, The TRA predicts health, voting, consumer spending, and religious activity. It predicts behavior better than models that solely examine attitudes, but it only applies to purposeful, controlled activity. When there are obstacles to participating in an activity, the idea of planned behavior must be utilized. Therefore, the TRA asserts that attitudes toward certain acts are based on expectations or perceptions about the expected outcomes [28]. Negative effects appear improbable if individuals feel good consequences are likely. Possibilities for beneficial outcomes are slim, and this is considered under beliefs and perceived behavioral control. Figure 1 below shows the interactions of elements implicit in the theory of reasoned action.

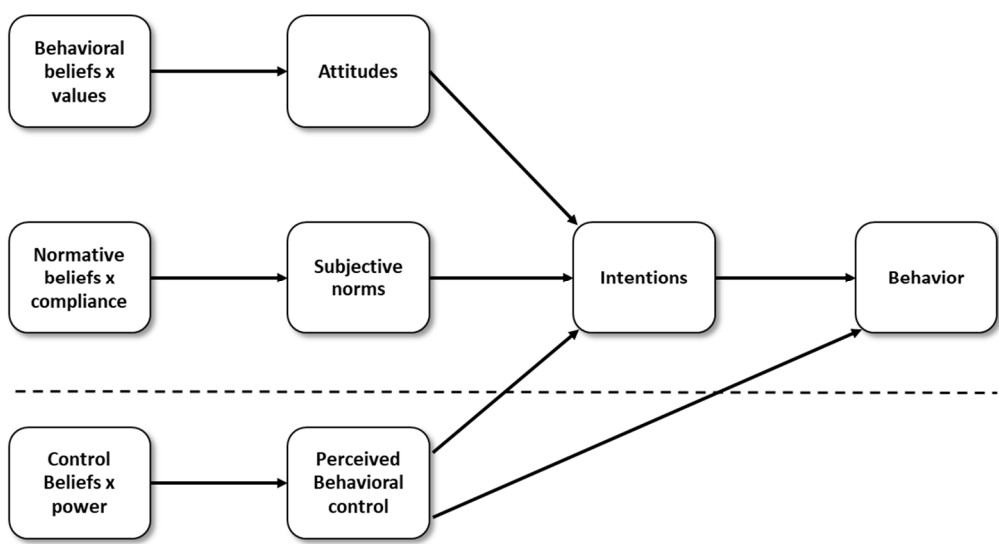

**Figure 1.** Illustration of the theory of reasoned action [29].

### 3.2. Theory of Planned Behavior

The theory of planned behavior (TPB) began as the theory of reasoned action in 1980 to forecast an individual's intention to participate in a behavior at a certain time and location. The hypothesis explains self-controllable behavior [30]. This model's fundamental component is behavioral intent, which is impacted by the probability of the anticipated result and the subjective judgment of the risks and rewards. The theory of planned behavior gives a better chance of understanding a person's true attitudes that result in physical behavior than does the theory of reasoned action [31]. The online interaction of children with apps may negatively influence their judgment because of the scenarios they observe. For example, if a child plays Grand Theft Auto and observes that stealing is rewarded, they may grow up with minor objections to corrupt-like behavior. Moreover, if the star players engage in socially uncouth habits, such as smoking and promiscuity, children may adopt these behavioral patterns in their adult lives. Figure 2 shows how the elements in the theory interact to result in an individual's ultimate behavior.

### 3.3. Conceptual Model

The study in focus proposes a conceptual model connecting game apps, social media apps, and video-streaming apps with children's behavior. The model also illustrates how artificial intelligence control and parental controls moderate the effect of games, social media, and video-streaming apps on children's behavior. Figure 3 below shows the hypothetical relationships between and among the study variables.

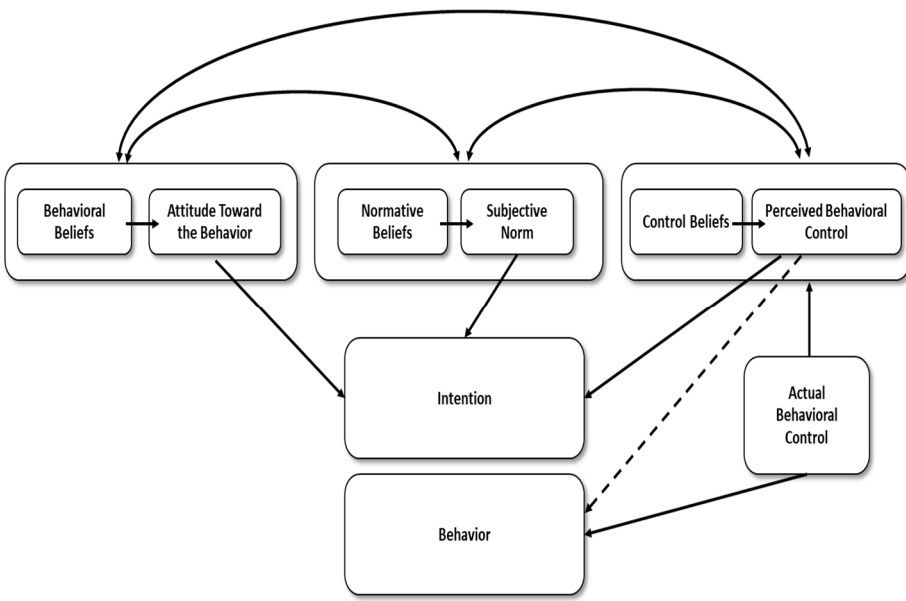

**Figure 2.** Illustration of the theory of planned behavior [32].

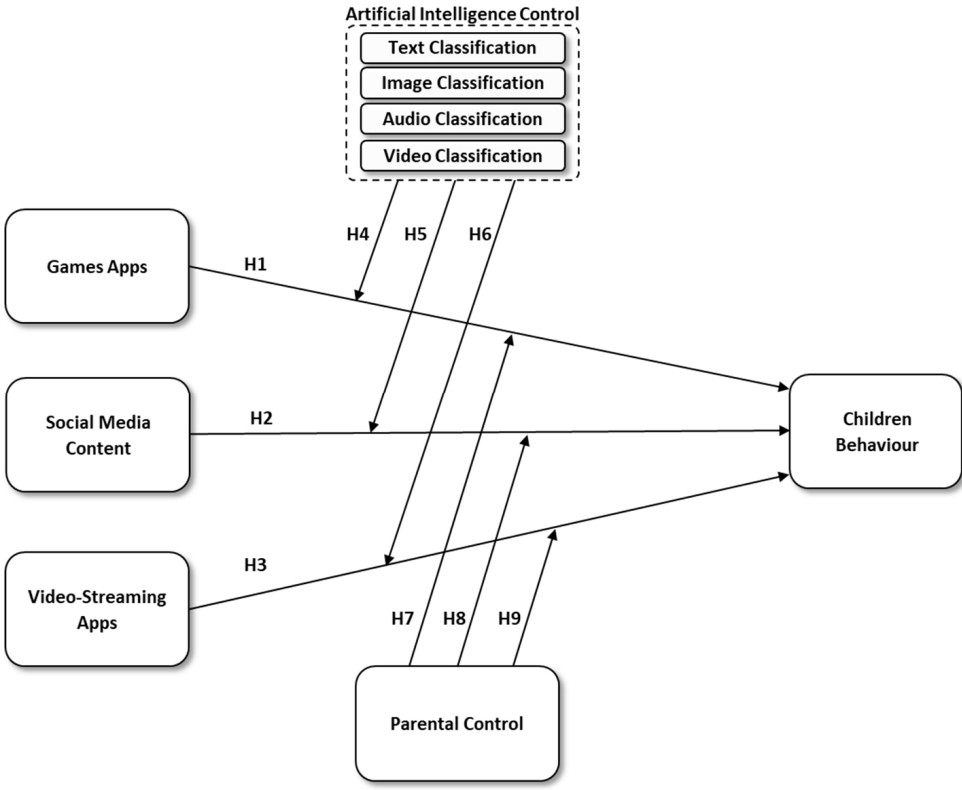

**Figure 3.** Conceptual framework of the study.

### *3.4. Hypotheses Development*

3.4.1. Games Apps on Child Behavior

In the gaming industry, most games meant for children feature violence. They occur in the form of a battle where a player shoots or stabs others to win [33]. Such a scenario builds the mentality in the children that they can use violence to get their way. Even the basic nature of games to demand that children should have some superpowers to win creates a false mindset in them as they grow. Those that consistently lose in the battles may grow up with low self-esteem. In the background, they are developing an unsustainable behavior

that may eventually manifest as depression or actual violent behavior against their peers. These games are likely to influence their development patterns, especially psychologically. For this reason, the researcher contended that:

**Hypothesis 1 (H1).** *Gaming apps have a significant negative effect on the sustainability of children's behavior.*

### 3.4.2. Social Media Apps on Child Behavior

Children's participation on social media should be minimal because even older persons suffer mental health issues from engaging in these platforms. Children are yet to establish resilience against body-shaming bullies and random disdainful remarks that others may hurl against them. Such remarks are likely to stick in their minds and cause depression because of their tender ages [34]. The platforms put pressure on users to emulate others while believing that they are not enough. A good example is Instagram, which is a platform on which users post their personal pictures and videos. These platforms cause mental health issues for children and teenagers who may want to look like popular and influential persons. Other platforms such as TikTok have a similar effect on pubescent children, where children compete to fulfill impossible challenges [35]. Unknowingly, it is a mirage, as there is no perfect state of being that will be acceptable to all opinions shared on social media sites. In this case, it will negatively influence their mental and psychological development, which is crucial in molding a child's behavior. For this reason, the researcher hypothesized that:

**Hypothesis 2 (H2).** *Social media apps have a significant negative effect on the sustainability of children's behavior.*

### 3.4.3. Video-Streaming Apps on Child Behavior

Video-streaming applications expose children to materials that may be age-inappropriate. Indeed, the platforms have regulatory mechanisms that prevent such exposure. However, taming a child's curiosity is difficult, especially if both parents are career people [36]. Children may start small by accessing child-specific content. With time, their curiosity will lead them to explore content that may disorient them from childhood thinking. Once the disorientation occurs, it may be an irreversible process that affects the child throughout childhood and youth. The mental growth of children is highly dependent on the movies and video materials they interact with the most. For this reason, the researcher hypothesized that:

**Hypothesis 3 (H3).** *Video-streaming apps have a significant negative effect on the sustainability of children's behavior.*

### 3.4.4. AI Control Moderating the Effect of Game Apps on Child Behavior

Integrating artificial intelligence into the games apps is good, as it adds to the quality of children's experience while they play. With this technology, players can enter their age details and create a child's profile. The AI algorithm is capable of filtering out inappropriate language if the player is a minor [37]. It can also use more appropriate images and textual or animated advertisements if it detects that the players are children. Most games meant for adults feature extreme violence and the use of insults. However, children may not realize it and proceed to play them, thereby exposing themselves to bad content and experiences. Artificial intelligence rids the systems of such an outcome by ensuring that all players engage in age-appropriate games. For this reason, the researcher contended that:

**Hypothesis 4 (H4).** *AI control positively moderates the effect of game apps on effect on the sustainability of children's behavior.*

### 3.4.5. AI Control Moderating the Effect of Social Media Apps on Child Behavior

By default, social media is not a friendly place for children because of the diversity of content posted on the networks. Artificial intelligence systems have attempted to solve this problem by creating kid-friendly social media platforms such as Kidzworld [38]. The platform actively evaluates material posted on the site for inappropriate content before approving it. It is an automated process that leverages machine learning tools and big data to ensure that the text, images, audio, and video media shared are on the platform. Parents can rest assured that their children do not expose themselves to bad content that may negatively influence their behavior. For this reason, the researcher hypothesized that:

**Hypothesis 5 (H5).** *AI control positively moderates the effect of social media apps on the sustainability of children's behavior.*

### 3.4.6. AI Control Moderating the Effect of Video-Streaming Apps on Child Behavior

The third hypothesis indicates the researcher's position that video-streaming applications negatively influence children's behavior sustainability. However, the integration of artificial intelligence is crucial to creating a safe environment for children on these platforms. The technology actively assesses the materials that a child searches and views. Based on this content, the AI can recommend age-appropriate videos. Another consideration that the algorithm takes as an input is a user's age, which one may declare [39]. Facebook, which doubles up as a video-streaming site, has integrated AI that detects graphic content such as accidents or nudity. It then blurs the content and cautions the user of the potentially disturbing video. Hence, the researcher hypothesized that:

**Hypothesis 6 (H6).** *AI control positively moderates the effect of video-streaming apps effect on the sustainability of children's behavior.*

### 3.4.7. Parental Control Moderating the Effect of Games Apps on Child Behavior

Parents' control over their children's exposure to inappropriate games is critical to developing sustainable behavior among children. Their physical presence and constant monitoring ensure that the children do not deviate from their course [40]. If left unsupervised, children can download other games that are not appropriate for their ages. The outcome is a child interacting with games that contain vulgar words and too much violence. Moreover, the parent's presence helps inexperienced children to obtain help on how to play, which may save them frustrating moments in their learning curve. This notion builds their self-confidence and molds good behavior in them. For this reason, the researcher hypothesized that:

**Hypothesis 7 (H7).** *Parental control positively moderates the effect of game apps effect on the sustainability of children's behavior.*

### 3.4.8. Parental Control Moderating the Effect of Social Media Apps on Child Behavior

Children's use of social media is a sensitive topic that parents dread. The image created by conventional social media platforms has made it impossible to convince some parents that some of these platforms are safe for children [41]. Those who accept ensure high-handedness in the activities that the children engage in while on social media. Some parents allow their children to have Facebook accounts provided they do not change their passwords. It ensures that the parent has ready access to the account to inspect the child's interactions. This mitigation approach helps create a safe online space for children to intermingle. By checking the conversation history and other posts, a parent can guide children on how to conduct themselves online. For this reason, the researcher hypothesized that:

**Hypothesis 8 (H8).** *Parental control positively moderates the effect of social media apps effect on the sustainability of children's behavior.*

3.4.9. Parental Control Moderating the Effect of Video-Streaming Apps on Child Behavior

Sometimes, a child may not notice that they are watching inappropriate content on video-streaming platforms. Some of these video clips have a mild connotation that gradually desensitizes children against odd behavior and languages [42]. They may be music videos, movies, or comedy shows. Parents can tell the subtle nature of these connotations and use this opportunity to guide their children away from these devious clips. Without this crucial supervision, children may become desensitized and indifferent to inappropriate dressing and language. Parents play a significant role in ensuring that children maintain a fair distance between them and the video content. For this reason, the researcher argued that:

**Hypothesis 9 (H9).** *Parental control positively moderates the effect of video-streaming apps effect on the sustainability of children's behavior.*

**4. Research Methodology**

*4.1. Item Measurement and Questionnaire Design*

The technique of research that was applied in the study was a quantitative approach. It is logical to go with this option when one considers that qualitative methods cannot be utilized to quantify the variables being studied [43]. In order to acquire the data, the researcher relied an online survey, which is concurrently characterized by efficiency, simplicity, and efficacy [44]. The primary questionnaire focused on game apps, social media apps, video-streaming apps, AI control, parental control, and child behavior. On the other hand, a separate questionnaire designed specifically to capture demographic information was also drafted. Appendix A shows the survey questions used in the study.

*4.2. Sampling and Data Collection*

An analysis of a population will often include taking a sample, which is a subset of the whole population. The study considered parents who had given their children smart devices and were willing to participate in the study in three selected towns, whose overall number was 1,616,755 [45]. Table 1 shows the breakdown of households allowing children to own smartphones in the eastern, western, and central regions of Saudi Arabia.

**Table 1.** Households with Children Owning Smartphones in Three Regions of Saudi Arabia.

| Region | Households |
|---|---|
| Eastern Region | 323,079 |
| Western Region | 658,094 |
| Central Region | 635,582 |
| Total | 1,616,755 |

The researcher relied on Yamane's formula to arrive at the conclusion that the minimum number of participants in the sample should be 400. The number served as the primary factor around which the analysis was predicated. The researcher chose which of the 400 possible responders would actually take part in the study by the use of a simple random sampling method. It is a kind of statistical sampling known as probabilistic sampling, and it involves providing each member with an opportunity to take part in a survey. It was determined to be appropriate due to the fact that every responder had almost identical traits with one another. The sole requirements for participation in the study were that applicants be willing to take part in the research and that they should have given their child or children a smart device.

The researcher utilized straightforward English in the design of the questionnaire to ensure that it would be understandable and used by all members of the group, even those

whose first language may not be English. In addition to this, it was translated into Arabic in order to better serve those individuals who have a limited understanding of English. The links were sent to 450 respondents, and 415 respondents duly filled the questionnaire. Hence, the response rate was 92.2%. Even more encouraging is the fact that each and every question was answered in full. This result was made possible by the researcher, who kept track of which participants had not responded to their questionnaire and sent out reminders to those individuals. Those who had only filled it in partly were also reminded of their obligation.

*4.3. Structural Equation Model*

The study adopted the structural model in analyzing the relationships between and among the constructs. In simulating the moderating role of artificial intelligence control and parental control, the researcher created other six variables:

- AI control × GM—Interaction between artificial intelligence control and game apps;
- AI control × SM—Interaction between artificial intelligence control and social media apps;
- AI control × VS—Interaction between artificial intelligence control and video-streaming apps;
- PC × GM—Interaction between parental controls and game apps;
- PC × SM—Interaction between parental controls and social media apps;
- PC × VS—Interaction between parental controls and video-streaming apps.

It also became necessary to assess the covariances between the purely independent variables. The same treatment was accorded to the variables in the AI-control-moderated and PC-moderated variables. The ultimate model is shown in Figure 4 below.

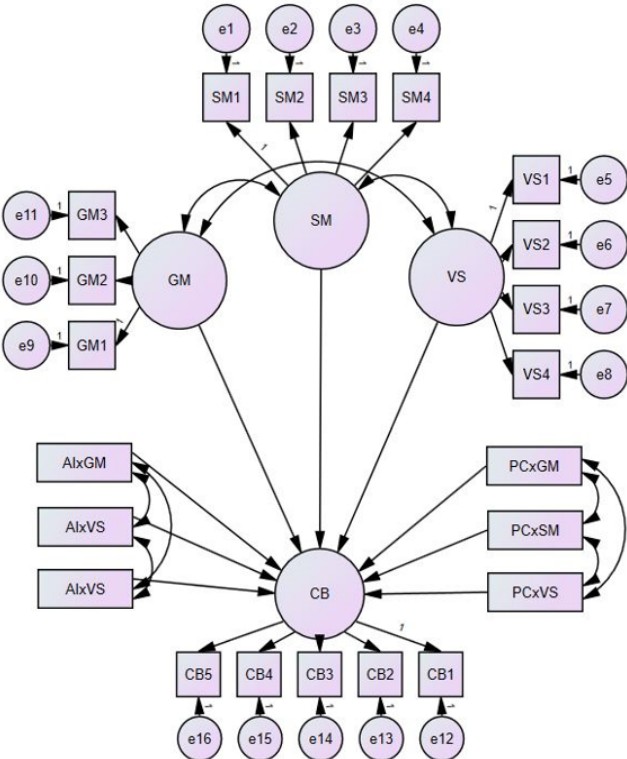

**Figure 4.** Structural Equation Model.

## 5. Data Analysis

*5.1. Analysis of Respondent Demographics*

The analysis of demographics considered the three parameters of gender, location, and number of children. With respect to gender, most respondents were female. This

demographic accounted for 57.1% of the respondents. Male participants were 42.9%. Location-wise, most of the respondents came from the central region of Saudi Arabia. They constituted 45.8%. Those coming from the west and east were 42.2% and 12.0%, respectively. Regarding the number of children, most of them had between four and five children, as this demographic constituted more than 50% of the sample. Those with only one child were 1.9%, representing the least dominant group. The above-mentioned results are summarized in Table 2.

**Table 2.** Demographic characteristics.

| Demographics | Frequency | Percentage (%) |
| --- | --- | --- |
| Gender | | |
|     Male | 178 | 42.9 |
|     Female | 237 | 57.1 |
| Location | | |
|     Eastern | 50 | 12.0 |
|     Western | 175 | 42.2 |
|     Central | 190 | 45.8 |
| Number of Children | | |
|     1 | 8 | 1.9 |
|     2 | 26 | 6.3 |
|     3 | 140 | 33.7 |
|     4 | 106 | 25.5 |
|     5 | 108 | 26.0 |
|     6 | 23 | 5.5 |
|     7 | 4 | 1.0 |

*5.2. Assessment of the Measurement Model*

5.2.1. Indicator Reliability

This statistic assesses the degree to which the variations of the question variables are comparable to those of their respective goal variables. A score of 0.7 or above is required as a bare minimum for approval [46]. The analysis that was carried out in this research showed that all of the construct variables scored values that were more significant than the necessary minimum value. The overall dependability score for the indicators was 0.873 on average. Table 3 summarizes these results.

5.2.2. Internal Consistency

The degree to which the component variables can explain the latent variables that have been allocated to them is what the internal consistency metric attempts to quantify. The statistic used to measure this metric was the Cronbach's alpha. The variables scored an average alpha of 0.786. The minimum required score is 0.70 [46]. Table 3 summarizes these results.

5.2.3. Convergent Validity

The average AVE scored to measure convergent validity was 0.652, as shown above in Table 3. The video-streaming apps variable had the lowest AVE score (0.600), while the child behavior variable received the greatest score (0.708). The minimum required score in convergent validity is 0.50 [47].

5.2.4. Discriminant Validity

A validity metric known as discriminant validity assesses whether or not questionnaire questions explain their target variable more than they explain other non-target variables. It is reasonable to anticipate that the answers pertaining to variable A will correlate strongly with variable A to a greater extent than they will correlate with variable B [47]. The instrument met the requirements for validity according to this criterion. All exogeneous variables

correlated more with their endogenous variables than they did with other variables. Table 4 shows the results obtained after establishing the validity scores.

**Table 3.** Construct indicators.

| Construct | Items | Factor Loading | Composite Reliability | Indicators | Cronbach's Alpha | AVE |
|---|---|---|---|---|---|---|
| Game Apps | GM 1<br>GM 2<br>GM 3 | 0.758<br>0.729<br>0.883 | 0.834 | 3 | 0.759 | 0.629 |
| Social Media Apps | SM 1<br>SM 2<br>SM 3<br>SM 4 | 0.788<br>0.841<br>0.809<br>0.775 | 0.879 | 4 | 0.772 | 0.646 |
| Video-Streaming Apps | VS 1<br>VS 2<br>VS 3<br>VS 4 | 0.854<br>0.677<br>0.699<br>0.850 | 0.856 | 4 | 0.806 | 0.600 |
| Child Behavior | CB 1<br>CB 2<br>CB 3<br>CB 4<br>CB 5 | 0.905<br>0.874<br>0.875<br>0.769<br>0.774 | 0.923 | 5 | 0.720 | 0.708 |
| Artificial Intelligence Control | AI 1<br>AI 2<br>AI 3<br>AI 4 | 0.756<br>0.851<br>0.881<br>0.796 | 0.893 | 4 | 0.874 | 0.676 |
| Parental Controls | PC 1<br>PC 2<br>PC 3 | 0.769<br>0.801<br>0.853 | 0.850 | 3 | 0.785 | 0.654 |

**Table 4.** Discriminant Validity Results.

| | GM | SM | VS | CB | AI | PC |
|---|---|---|---|---|---|---|
| GM | 0.8472 | | | | | |
| SM | 0.4725 | 0.7625 | | | | |
| VS | 0.4638 | 0.4783 | 0.8477 | | | |
| CB | 0.4066 | 0.4508 | 0.5205 | 0.753 | | |
| AI | 0.4200 | 0.4586 | 0.5013 | 0.470 | 0.7519 | |
| PC | 0.4638 | 0.4455 | 0.4663 | 0.443 | 0.5014 | 0.8286 |

*5.3. Model Fit Indices*

5.3.1. Fit Indices for the Structural Model

Results from the model fit indices show that the outcome of the structural model is statistically significant. The chi-square score of the model is 705.4074 (df = 197, $p$ = 0.000). It provides evidence that the model had a major impact on the statistical analysis. The CMIN/DF score of the default model is 3.581. According to [48], if this score is below 5, the model indicates a reasonable fit. The results of the default and independence models are summarized in Table 5. These models all point to the same conclusion.

**Table 5.** Fit Indices for the Structural Model.

| Model | CMIN | DF | P | CMIN/DF |
|---|---|---|---|---|
| Default model | 705.4074 | 197 | 0.000 | 3.581 |
| Independence model | 898.8569 | 231 | 0.000 | 3.891 |

CMIN, chi-square value; DF, degree of freedom.

### 5.3.2. The Effect of Smart Devices Applications on Child Behavior

In this section, the goal was to establish the effect of children's exposure to the three kinds of smart devices applications on their behavior. Table 6 shows the regression statistics obtained from the computation.

**Table 6.** Regression Smart Devices Applications × Child Behavior.

| Summary Stats | | | | |
|---|---|---|---|---|
| **R-Squared** | **Adj. R-Squared** | **F Score** | **Sig/*p*-value** | |
| 0.447 | 0.443 | 110.596 | 0.000 | |
| **Coefficients** | | | | |
| **Variable** | **Beta** | ***t*-stat** | ***p*-value** | **Decision** |
| Constant | 5.414 | 50.982 | 0.000 | Supported |
| Games Apps | −0.195 | −4.191 | 0.000 | Supported |
| Social Media Apps | −0.409 | −8.334 | 0.000 | Supported |
| Video-Streaming Apps | −0.239 | −4.874 | 0.000 | Supported |

The study showed that the r-square coefficient was 0.447 (F = 110.596, $p$ = 0.000), which implies that the effect of smart device applications on a child's behavior is significant [49]. The r-squared illustrates that 44.7% of the variation in child behavior could be explained by the variation in the three kinds of smart devices applications, namely game apps, social media apps, and video-streaming apps. All applications contributed negatively to children's behavior, as they all had negative t and beta values. Independent variables with higher beta and t-scores denote higher effect than those with lower values of the same [49]. The applications with the biggest effect were social media applications, as they scored a beta of −0.409 (t = −8.334, $p$ = 0.000).

### 5.3.3. The Moderating Effect of AI control on the Effect of Smart Device Applications on Child Behavior

In this part, the goal was to establish how artificial intelligence control moderates the effect of children's exposure to the three kinds of smart device applications on their behavior. Table 7 shows the regression statistics obtained from the computation.

**Table 7.** Regression AI Control Moderating Effect.

| Summary Stats | | | | |
|---|---|---|---|---|
| **R-Squared** | **Adj. R-Squared** | **F Score** | **Sig/*p*-value** | |
| 0.152 | 0.146 | 24.643 | 0.000 | |
| **Coefficients** | | | | |
| **Variable** | **Beta** | ***t*-stat** | ***p*-value** | **Decision** |
| Constant | 2.223 | 13.682 | 0.000 | Supported |
| AI control × Games Apps | 0.070 | 4.255 | 0.000 | Supported |
| AI control × Social Media Apps | 0.071 | 4.184 | 0.000 | Supported |
| AI control × Video-Streaming Apps | 0.034 | 1.889 | 0.010 | Supported |

The study showed that the r-square coefficient was 0.152 (F = 24.643, $p$ = 0.000), which implies that the AI control moderating effect on the influence of smart devices applications on child's behavior is significant [49]. The r-squared illustrates that 15.2% of the variation in child behavior could be explained by the variation in the use of AI on the respective smart device applications. All moderating variable pairs contributed positively to children's behavior, as they all had positive t and beta values. The combination with the biggest effect was AI control x social media apps, as it scored a beta of 0.071 (t = 4.184, $p$ = 0.000).

### 5.3.4. The Moderating Effect of PC on the Effect of Smart Devices Applications on Child Behavior

The goal in this section was to establish how parental controls moderate the effect of children's exposure to the three kinds of smart device applications on their behavior. Table 8 shows the regression statistics obtained from the computation.

**Table 8.** Regression Parental Control (PC) Moderating Effect.

| Summary Stats | | | |
|---|---|---|---|
| **R-Squared** | **Adj. R-Squared** | **F Score** | **Sig/*p*-value** |
| 0.067 | 0.060 | 9.761 | 0.000 |
| **Coefficients** | | | |
| **Variable** | **Beta** | ***t*-stat** | ***p*-value** | **Decision** |
| Constant | 2.832 | 19.454 | 0.000 | Supported |
| PC × Games Apps | 0.050 | 2.790 | 0.006 | Supported |
| PC × Social Media Apps | 0.048 | 2.611 | 0.009 | Supported |
| PC × Video-Streaming Apps | 0.002 | 0.085 | 0.932 | Supported |

The study showed that the r-square coefficient was 0.067 (F = 9.761, $p = 0.000$), which implies that the parental control (PC) moderating effect on the influence of smart applications devices on child's behavior is statistically significant [49]. The r-squared illustrates that 6.7% of the variation in child behavior could be explained by the variation in the use of PC in the respective smart devices' applications. All moderating variable pairs contributed positively to children's behavior, as they all had positive t and beta values. The combination with the biggest effect was PC × game apps, as it scored a beta of 0.050 (t = 2.790, $p = 0.000$). The application of parental controls on video-streaming apps was statistically insignificant, as it scored a beta coefficient of 0.002 (t = 0.085, $p = 0.932$) [50].

## 6. Discussion, Implications, and Limitations

### 6.1. Discussions of Findings

Game applications were significant in negatively influencing sustainable child behavior among Saudi children. This effect is consistent with [51], as the cited study observes that children introduced to violent games tend to exhibit mild to extreme violent behavior in their childhood and later stages in life. The logic behind it is that the child adopts the conduct of the characters they see on the games. Furthermore, most games celebrate violence and normalize it. Protagonists in such games engage in violence to obtain rewards, which a child can easily extrapolate to real-life situations. Nevertheless, the study by [52] argues that violence in games does not significantly mold unsustainable violent behavior among children. This investigation findings suggest that allowing children to have uncontrolled access to game applications on smart devices is not advisable.

The use of social media applications among Saudi children significantly affects the sustainability of children's behavior negatively. The more a child interacts with social media, the more likely they are to encounter inappropriate posts. Their minds are still in their development stages and cannot properly process such information. According to [17], a single encounter with inappropriate material on social media can significantly disorient them for a long time. Some become addicted to viewing such posts because of the excitement that comes with the experience. In the absence of proper controls, a child can end up interacting with the wrong people and content on these platforms. The overall suggestion made in this study is that children's uncontrolled use of social media can prove disastrous to the sustainability of their behavior.

The study found video streaming platforms to have a detrimental effect on the sustainability of children's behavior in Saudi Arabia. Children visiting these sites are likely to encounter material that is not appropriate for their ages. For example, a child on the larger YouTube platform can see and interact with songs that show romantically involved

persons engaging in sexual activities. Because they are not used to this content, it may capture their attention, which may lead to them searching for more such videos. The study by [20] argued that YouTube and Netflix deprive children of their childhood growth by exposing them to material that is not suitable for child viewership. These findings suggest that children's uncontrolled access to video-streaming platforms is not healthy for the sustainability of their behavioral development.

The study investigated the moderating effect of artificial intelligence control between the three kinds of smart devices' applications and the sustainability of children's behavior. Findings indicated that when combined with AI, game, social media, and video-streaming apps have a significant positive effect on child behavior sustainability. Its mechanisms of filtering content by text, audio, video, and images seem to be effective in ensuring that children do not come into direct contact with undesirable content. The study by [22] argues that the models used in achieving this goal, such as K-nearest neighbor (KNN), decision trees, and support vector machines (SVM), are very effective in classifying content based on selected features. These algorithms are good at learning users' behaviors and can detect if the user is a child or an adult based on recent searches and watch history. Their automated nature implies that parents can rely on them to proof content on games, social media, and video-streaming platforms even without the parents' direct involvement.

Findings also established that parental controls are significantly effective in proofing content for the children's consumption. The implication is that parents are capable of determining which content is suitable and appropriate for children. Nevertheless, these controls were not found to be as effective as AI-based controls. One of the reasons that led to their poor performance was that parents were not there all the time to monitor their children [26]. When parents are away, children could wander off into the abyss of unregulated content. Additionally, it is difficult for parents to know about every other piece of content on the platforms. Therefore, they have a limited ability to detect inappropriate content. For example, a YouTube thumbnail may look innocent, while its content has inappropriate footage. For this reason, the study suggests that parental controls have limited effectiveness in moderating the effect of smart devices' applications on the sustainability of child behavior.

### *6.2. Implications*

From the discussion of study findings, it is evident that game, social media, and video-streaming applications have a negative effect on the sustainability of child behavior. If children are left to consume content and interact with these applications, their behavior is likely to plummet into oblivion. The study has established that artificial intelligence control plays a crucial moderating role in averting the negative effects of the smart devices' applications on child behavior sustainability. They have an automated system that categorizes content based on its features. As a result, the algorithms not only recommend appropriate content but also correctly distinguish a child from an adult user. Parental controls are good at proofing content, but their reliability is in question. Therefore, if parents wish to ensure minimal negative effects from their children's interactions with smart devices' applications, they need to allow them to engage with platforms that have machine learning algorithms targeted at classifying content for children's consumption.

### *6.3. Limitations*

The biggest limitation of this study is that it focused on parents of the children instead of the children. The actual people negatively affected by the games, social media, and videos are the children. However, due to ethical issues, the researcher could not directly engage with the children. Nevertheless, the parents provided a great deal of information to simulate the experiences their children are going through while interacting with the smart devices' applications. Another limitation is that the study could not investigate the long-term effects of the applications on child behavior. Such results can only come from a longitudinal study, while the current one was cross-sectional. Thirdly, the parents may

have been unnecessarily defensive when asked about their children's behavior. They may have found it not beneficial to portray their children in bad light.

## 7. Conclusions and Future Research

This study has established that allowing children unregulated access to games, social media, and video-streaming applications on their smart devices is unhealthy for the sustainability of their behavior. Such children tend to exhibit conduct that becomes socially unfavorable. The reason justifying this outcome is the inappropriate material they keep interacting with while on these platforms. Parents have two options to exercise in bringing their children's behavioral sustainability to normalcy. Firstly, they should ensure they have parental controls on the gadgets and monitor their children's online activities. Secondly, and even more effectively, parents should only allow their children on platforms that have applied artificial intelligence control in classifying their content. In this case, the children will only be exposed to age-appropriate content. Their behavior is not likely to deteriorate in this setting. Future research should focus on the long-term effects of children's interaction with smart-devices' applications. Instead of using cross-sectional methods, the studies should consider longitudinal approaches in which they can monitor children's behavioral trends over time.

**Author Contributions:** Conceptualization, O.A.; Formal analysis, O.A.; Investigation, O.A.; Methodology, H.B.; Validation, H.B.; Writing—review & editing, H.B. All authors have read and agreed to the published version of the manuscript.

**Funding:** This research received no external funding.

**Institutional Review Board Statement:** Not applicable.

**Informed Consent Statement:** Not applicable.

**Data Availability Statement:** The data presented in this study are available on request from the corresponding author.

**Conflicts of Interest:** The authors declare no conflict of interest.

## Appendix A.

**Table A1.** The Questions Used in the Survey.

| Construct | Questions | Inspiring Source |
|---|---|---|
| Game Apps | Your children are fond of playing games on smart devices | [53] |
| | The games installed on your child's device are very engaging | [53] |
| | Your child participates in online gaming competitions | [53] |
| Social Media Apps | Your child has access to social media sites on their device | [18] |
| | Your child posts on social media sites | [18] |
| | Your child finds it enjoyable to be on social media | [18] |
| | Your child participates in social media challenges | [18] |
| Video-Streaming Apps | You have observed your child go on YouTube or Netflix | [54] |
| | Your child finds it enjoyable to be on video-streaming applications | [54] |
| | You have found your child posting on YouTube | [54] |
| | Your child has some ideas of being a video content creator | [54] |
| AI Control | The textual content on your child's smart devices is age-appropriate | [24] |
| | The videos on streaming sites visited by your child are appropriate | [24] |
| | The music and songs played on the applications are appropriate for children | [24] |
| | The images your child sees on their applications are appropriate for their age | [24] |
| Parental Control | You closely monitor your child's online activity | [55] |
| | Sometimes, you play games and watch online videos with your child | [55] |
| | You actively advise your child against bad content and games | [55] |
| Child Behavior | Your child is generally obedient to adults | [56] |
| | Your child is highly disciplined | [56] |
| | Your child does not get into fights with others | [56] |
| | Your child is morally upright | [56] |
| | Your child does not have bad friends | [56] |

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
