# Peer review of "The Sustainable Effect of Artificial Intelligence and Parental Control on Children’s Behavior While Using Smart Devices’ Apps: The Case of Saudi Arabia"

_sustainability, doi:10.3390/su14159388_

Round 1
Reviewer 1 Report
Questions:
- What do you mean by sustainable behavior and sustainable development?
- The control component of the TRA was not as well expounded.
- How did the TRA and TPB lead to the conceptual framework used in this research?
- What is currently researched about parental controls/AI controls in prepubescent usage? What is new in this research in contrast to other similar research?
Small sugestions:
- The first paragraph of the Literature Review covers two different topics: children being exposed to serious games for positive development, and games that may lead to violent behavior though not strongly proven. Consider breaking it down into separate paragraphs. I think this was done because you intended to just have a single paragraph for each theme to be covered, but it is not necessary to economize in this way. Consider other paragraphs in the Literature Review, too.
- Consider more thorough proofreading. In several cases, the subject "researchers" is paired with a singular verb.
- Please check Hypothesis 7 (does not match the subsection it is in)
- Section 1.3: Define the acronyms on first use
Reviewer 2 Report
The study is quite interesting and innovative.
It is stated that "children now spend between one and two hours on their mobile devices". What ages are this children? this sentence needs to be supported by bibliography: "At this age, they are primarily still in their 41 prepubescent ages. Their exposure to these gadgets at an early age has been a subject of 42 concern to their parents, who feel that they cannot handle the nature of some online material. Their specific fear is that their children will come into contact with some age-inappropriate content, which is not good for the sustainable development of their mental 45 health." The same here: "Some even argue that this technology further facilitates educational progress by exposing children to volumes of valuable materials. As intimated in 62 the findings above, scholarly opinion is split between allowing children limited access to 63 mobile devices and completely disallowing this access. The issue with disallowing access 64 is that the children would find other means of accessing the materials all the same. It is far 65 more dangerous because parents would not be in control of this access, and the material 66 could be more inappropriate. Children often model their behavior from the materials they 67 interact with online, and it is an unsustainable practice for them to have unlimited access 68 to online materials." The same here: "This unsupervised access to limitless online materials, games, and 82 content may result in the unsustainable development of children’s mental and psychological health."Author Response
Please check the attached file
